# Experimental Selection of Paromomycin Resistance in *Leishmania donovani* Amastigotes Induces Variable Genomic Polymorphisms

**DOI:** 10.3390/microorganisms9081546

**Published:** 2021-07-21

**Authors:** Sarah Hendrickx, João Luís Reis-Cunha, Sarah Forrester, Daniel C. Jeffares, Guy Caljon

**Affiliations:** 1Laboratory of Microbiology, Parasitology and Hygiene (LMPH), University of Antwerp, 2610 Antwerp, Belgium; sarah.hendrickx@uantwerpen.be; 2Department of Biology and York Biomedical Research Institute, University of York, York YO31 5DD, UK; jaumlrc@gmail.com (J.L.R.-C.); sarah.forrester@york.ac.uk (S.F.)

**Keywords:** paromomycin, resistance, sequencing, SNP, CNV

## Abstract

The relatively high post-treatment relapse rates of paromomycin (PMM) in visceral leishmaniasis treatment and the swift emergence of experimental drug resistance challenge its broad application and urge for rational use and monitoring of resistance. However, no causal molecular mechanisms to *Leishmania* PMM resistance have been identified so far. To gain insights into potential resistance mechanisms, twelve experimentally selected *Leishmania donovani* clonal lines and the non-cloned preselection population, with variable degrees of PMM resistance, were subjected to whole genome sequencing. To identify genomic variations potentially associated with resistance, SNPs, Indels, chromosomal somy and gene copy number variations were compared between the different parasite lines. A total of 11 short nucleotide variations and the copy number alterations in 39 genes were correlated to PMM resistance. Some of the identified genes are involved in transcription, translation and protein turn-over (*transcription elongation factor-like protein*, *RNA-binding protein, ribosomal protein L1a, 60S ribosomal protein L6, eukaryotic translation initiation factor 4E-1, proteasome regulatory non-ATP-ase subunit 3*), virulence (*major surface protease gp63, protein-tyrosine phosphatase 1-like protein*), mitochondrial function (*ADP/ATP mitochondrial carrier-like protein*), signaling (*phosphatidylinositol 3-related kinase*, *protein kinase putative* and *protein-tyrosine phosphatase 1-like protein*) and vesicular trafficking (*ras-related protein RAB1*). These results indicate that, in *Leishmania*, the aminoglycoside PMM affects protein translational processes and underlines the complex and probably multifactorial origin of resistance.

## 1. Introduction

*Leishmania* parasites belong to the family of Trypanosomatidae and are unicellular flagellates that are transmitted between different vertebrates via the bites of infected sand flies. Depending on the species, *Leishmania* can cause either cutaneous (CL), mucocutaneous (MCL) or visceral leishmaniasis (VL). These parasites are known for their high plasticity and rapid adaptation to different environmental conditions, such as drug exposure [1].

The efficacy of the aminoglycoside antibiotic paromomycin (PMM) against both CL and VL has been demonstrated in monotherapy and in combination with other anti-leishmanials, with cure rates up to 80% [2,3,4,5,6]. The relatively high post-treatment relapse rates challenge the broad application of PMM in VL treatment, and urge for rational use and monitoring of the emergence of resistance [7], particularly because several in vitro and in vivo laboratory studies demonstrated fairly easy selection of resistance both on promastigotes and intracellular amastigotes [8,9,10,11]. A stepwise increase of PMM pressure onto promastigotes led to a decreased PMM susceptibility of both stages, whereas exposure of intracellular amastigotes in vitro or in vivo resulted in an amastigote-specific increase of PMM IC_50_ [9]. This stage-dependent outcome disfavors the use of the promastigote stage for primary resistance monitoring and emphasizes the need for resistance markers to monitor the emergence and spread of PMM resistance in the field [9]. With the aim to discover putative PMM resistance markers and molecular mechanisms enrolled in PMM resistance, whole genome sequences of experimentally amastigote-selected clones with different levels of PMM susceptibility were compared among each other and to the wild-type (WT) drug-susceptible parent strain. This led to the identification of short-nucleotide variants (SNVs) and gene expansion/deletions that could be associated with PMM resistance.

## 2. Materials and Methods

Promastigotes of *L. donovani* MHOM/NP/03/BPK275/0-cl18, isolated within the frame of the EC Kaladrug-R project from a patient not responding to treatment with sodium stibogluconate, were sub-cultured twice weekly in HOMEM promastigote medium (Life technologies, Ghent, Belgium) at 25 °C. Resistance was selected in vitro on intracellular amastigotes, as previously described [9,12]. Twelve clonal isolates were obtained from the resulting polyclonal-resistant population (Table 1). The PMM and miltefosine (MIL) susceptibility of intracellular amastigotes was determined for each clone and the non-cloned population, as described previously [8]. In addition, their susceptibility to amphotericin B (AmB; Sigma Aldrich, Overijse, Belgium) was calculated to evaluate potential cross-resistance.

Promastigotes from twelve clones and the non-cloned population in exponential growth phase were used for DNA extraction by the QIAamp DNA Mini Kit according to the manufacturer’s instructions (Qiagen, Antwerp, Belgium). Genomic DNA was used to generate paired-end libraries using the NEBNext Ultra II FS DNA library prep kit for Illumina sequencing (New England Biolabs, Ipswich, MA, USA), according to the manufacturer’s protocol, and amplified using NEBNext multiplex oligos for Illumina (dual index primers, New England Biolabs). Pooled and barcoded libraries were sequenced to 2 × 150 bp on an Illumina Hiseq 3000 platform. Reads were mapped to the *L. donovani* BPK282A1 genome version 39 available from TriTrypDB using BWA mem [13,14]. The genome coverage was estimated based on the mean read depth of genes with non-outlier depth coverages for each isolate, using BEDtools genomecov version 2.27.1 (BEDTools, version 2.27.1; A suite of utilities for comparing genomic features; Aaron Quinlan and Ira Hall; Bioinformatics: University of Oxford, Oxford, England 2010) [15] and GFF coordinates. Detailed descriptions of each read library are summarized in Appendix A.

Short-nucleotide variant (SNV) distribution and allele frequency variation among *L. donovani* isolates were used to estimate populational diversity and potential association of mutations with PMM resistance. SNVs were identified with a focus on single-nucleotide polymorphism (SNPs) and short insertion–deletion mutations (INDELS). Calls for each sample were generated with FreeBayes version 1.2.0 (FreeBayes, version 1.2.0; Haplotype-based variant detection from short-read sequencing; Eric Garrison and Gabor Marth; arXiv:1207.3907; 2012) [16], calling only on positions where the depth was at least 5 reads. SNVs were filtered by: quality > 2000, minimum global allele frequency (MAF) > 0.05, number of different alleles = 1, mapping quality and base quality of the reference and alternate allele > 40, and proportion of observed reference and alternate alleles which were supported by properly paired read fragments > 0.9, using VCFtools version 0.1.16 (VCFtools, version 0.1.16; The variant call format and VCFtools; Adam Auton, Petr Danecek and Anthony Marcketta; Bioinformatics: University of Oxford, Oxford, England 2011) [17] and BCFtools version 1.10.2 (BCFtools, version 1.10.2; Utilities for variant calling and manipulation VCFs and BCFs; Heng Li; Bioinformatics: University of Oxford, Oxford, England 2011) [18]. To identify SNVs that occurred during selection for PMM resistance, from the total of 7448 SNVs, we excluded the 5400 variants that presented MAF < 0.05 or low-quality calls, representing unreliable calls or variants that were present in all isolates, as these are simply differences between the isolates and the reference assembly. The sharing of SNV positions among the 13 isolates was estimated using BCFtools isec, and plotted in R using ggplot2. The populational genetic diversity (π) was estimated with VCFtools window-pi, in 50 kb windows. The images of the diversity values of these windows along the 36 *L. donovani* chromosomes were generated in R, using ggplot2 (https://cran.r-project.org/web/packages/ggplot2/index.html; accessed on 19 July 2021).

To estimate fluctuation in the ‘within-clone allele frequency’ for each SNV, the frequency of the alternate allele was estimated by dividing the read depth of the alternate allele by the total number of reads that mapped in the position. The heatmap representing the fluctuations in alternate allele frequency along the chromosomes was generated in R, with the heatmap2 function, from the gplots library (https://cran.r-project.org/web/packages/gplots/index.html; accessed on 19 July 2021). *L. donovani* isolates were clustered based on the Manhattan distance of the allele frequency of all SNVs, by UPGMA (unweighted pair group method with arithmetic mean). SNV positions were drawn according to their chromosomal positions, or reordered based on clustering by UPGMA of their Manhattan distance. The genome-wide distribution density of the alternate allele frequencies of each *L. donovani* sequenced isolate was estimated in R, with the density function, and used to infer the parasite ploidy.

The potential impact of fluctuation in allele frequency of each SNV with PMM resistance was estimated by the Pearson correlation of the allele frequency and the isolate PMM IC_50_, in R, and the images were generated with ggplot2.

To evaluate the impact of gene and chromosomal copy alterations on PMM resistance, the copies of all *L. donovani* 8135 genes and 36 chromosomes were estimated for the 12 sequenced clonal lines and the non-cloned population, and compared with their PMM resistance profiles. To estimate gene copy numbers, the median read depth of each gene was calculated with BEDtools genomecov version 2.27.1 (BEDTools, version 2.27.1; a suite of utilities for comparing genomic features; Aaron Quinlan and Ira Hall; Bioinformatics: University of Oxford, Oxford, England 2010), using gene coordinates from *L. donovani* GFF version 39 from TriTrypDB. Read depths were transformed to relative copy number values (copies per haploid genome copies) by dividing the gene coverages by the genome coverage, estimated as the median coverage of non-outlier genes (Grubb’s tests, with *p* < 0.05). The somy of each chromosome from each *L. donovani* clone was estimated based in the median coverage of non-outlier genes, as described by Reis-Cunha et al. [19]. Briefly, mapped reads were filtered by mapping quality 30 using SAMtools version 1.10 (SAMtools, version 1.10; Suite of programs for interacting with high-throughput sequencing data; Heng Li; Bioinformatics: University of Oxford, Oxford, England 2009) [20] and genes with outlier coverages were excluded, based on iterative Grubb’s tests, with *p* < 0.05. The median read depth of all non-outlier genes in each chromosome was normalized by the genome coverage and considered as the chromosomal somy. The potential impact of each *L. donovani* gene copy number and chromosome somies on PMM resistance was estimated by Pearson correlation of the gene copy number or somy against clone PMM IC_50_ values. All statistical analyses were carried out using R v3.6.2 (R, version 3.6.2; The R project for Statistical Computing; R core team; Vienna, Austria, 2019) [21], and graphical representations were generated with ggplot2 [22].

## 3. Results

### 3.1. Amastigote Susceptibilty to Paromomycin and Amphotericin B

As described elsewhere [8], all in vitro selected clones showed lower susceptibility to PMM than the parent strain, in several cases achieving 5–10-fold IC_50_ values (Table 1). To evaluate potential cross-resistance to other drugs, the intracellular drug susceptibility to amphotericin B was determined (Table 1). The in vitro susceptibility of this naturally antimony-resistant strain against pentavalent and trivalent antimonials remained unaltered, as did the susceptibility against MIL [9]. PMM and amphotericin B susceptibility were not significantly correlated, which reinforces that the developed PMM resistance is specific to this drug, and not a result of generic drug resistance mechanisms.

### 3.2. SNPs and Indels

We detected 7448 single-nucleotide variants (SNV), where 2048 (1222 single-nucleotide polymorphisms (SNP) and 826 indel positions) vary among the evaluated *Leishmania* isolates (Appendix A). Overall, the majority of these positions were present in all post-selection clones (Figure 1a,b), which is expected given their similar origin. When looking at the genetic diversity (π) between the different clones and the pre-selection sample (Appendix A), chromosomes 6, 12 and 34 (mostly euploid) and chromosome 31 (aneuploid) had relatively high π spikes, which were generally a result of co-localization of SNPs and indels (Figure 1c). Even though the SNV positions were generally shared amongst most post-selection clones, variations in within-clone allele frequency patterns were observed (Figure 2). These inconsistent counts of reads from the two alleles may be due to gene duplications/losses. Therefore, SNVs in these clones were further compared at the allele frequency level in the downstream analyses. It is noteworthy that the genome-wide within-clone allele frequency patterns suggest that the progenitor isolate was triploid, as are all post-selection clones. We expect that the variation in within-clone allele frequencies is due to loss of heterozygosity due to fluctuating ploidy [23,24] and/or the difficulty in estimating true allele counts in a triploid sample.

To identify SNVs that associate with PMM resistance, we have compared within-clone allele frequencies to PMM IC_50_ values. A total of 11 SNV positions were correlated with PMM resistance, from which 9 were located in intergenic regions and 2 resulted in synonymous mutations (Figure 3, Appendix A). These SNVs were not clustered in close chromosomal regions and were dispersed over different regions and chromosomes. The intergenic SNPs on chromosomes 18 and 35 showed the strongest correlations with resistance and are located upstream of the *chaperone DnaJ protein* (LdBPK_181410.1) and *aspartate aminotransferase* genes (LdBPK_350840.1). The 2 SNPs resulting in synonymous mutations were found in genes LdBPK_020680.1 (*ATP-dependent Clp protease subunit, heat-shock protein 78* (*HSP78*)) and LdBPK_362510.1 (*sterol 24-c-methyltransferase*) (Appendix A). For both genes, a negative correlation was found between the alternate allele frequency and PMM resistance, suggesting that loss of the alternate alleles towards a reference-homozygous genotype correlates with PMM resistance. The synonymous SNP in LdBPK_020680.1 (UCG → UCC) results in a lower relative synonymous codon usage of the alternate codon (RSCU decrease from 1.439 to 1.115). For the *sterol 24-c-methyltransferase*, the synonymous mutation results in a higher frequency codon (UUU → UUC) with a substantial RSCU increase from 0.72 to 1.28 [25].

### 3.3. Chromosomal Copy Number Variation (CCNV)

Of the 36 *L. donovani* chromosomes, two had significant correlations between chromosome expansion and PMM resistance, Chr33 (r = 0.64, *p*-value = 0.02) and Chr34 (r = 0.56, *p*-value = 0.04) (Appendix A). However, the correlation, r, was lower than 0.7 and the amplitude of variation in these chromosome copies was low (Chr33, minimum = 1.29, maximum = 1.39; Chr34, minimum = 0.89, maximum = 0.94). Within their complex pattern of CCNV, the “chromosomal copy number drift” that could be observed for all of the clones (Appendix A) might be potentially explained by adaptation to in vitro culture [26,27].

### 3.4. Gene Copy Number Variations (CNV)

A significant correlation could be noted between PMM resistance and an increased or decreased gene copy number of 33 and 6 genes, respectively (Figure 4, Appendix A). The genes with the highest correlation to PMM resistance are summarized in Table 2. The gene with the largest difference between the maximum and minimum gene copy values is a small hypothetical protein (LdBPK_020370.1), with no identifiable functional motifs. Although 14 genes are hypothetical genes, the other 25 are putatively enrolled in several functions, such as: transcription, translation and protein turn-over (*transcription elongation factor-like protein, RNA-binding protein, ribosomal protein L1a, 60S ribosomal protein L6, eukaryotic translation initiation factor 4E-1, proteasome regulatory non-ATP-ase subunit 3* and *RING-H2 zinc finger,* with possible involvement in protein ubiquitination), virulence (*major surface protease gp63, protein-tyrosine phosphatase 1-like protein*), mitochondrial function (*ADP/ATP mitochondrial carrier-like protein*), signaling (*phosphatidylinositol 3-related kinase*, *protein kinase putative* and *protein-tyrosine phosphatase 1-like protein*) and vesicular trafficking (*ras-related protein RAB1*).

## 4. Discussion

Although PMM resistance mechanisms are well-established for bacteria [28,29], several hypotheses have been postulated for *Leishmania*, such as interference with ribosomal protein synthesis and inhibition of respiration [11,30,31,32,33,34,35]. However, no genetic resistance markers have been identified as yet. To try to identify these markers, the present study performed whole genome sequencing and comparison of 12 amastigote-selected clones and the pre-selection population, that had a large range of PMM resistances (Table 1). Consistent with previous analyses, we were not able to identify a specific genetic change that unequivocally explains PMM resistance. While the 12 post-selection clones contained numerous new SNV mutations that were fixed in all post-selection clones, these common mutations cannot explain the large variation in PMM resistance within these clones. As clone IC_50_ values range from 57 µM (only marginally greater than the pre-selection strain at 45 µM) to 417 µM, we hypothesized that multiple other non-fixed genetic differences contribute to PMM resistance. To determine which of these genetic differences may be associated with resistance, we compared within-clone allele frequencies, gene CNV and chromosome CNV with PMM resistance, using a similar approach to the genome-wide association study (GWAS) approach.

The majority of the SNV positions whose within-clone allele frequency correlated with PMM resistance were observed in intergenic regions, including notably two upstream of the *chaperone DnaJ protein* and *aspartate aminotransferase* genes. Aspartate aminotransferase has been described to be significantly elevated in patients receiving PMM, indicating hepatotoxicity or another drug effect [36]. Further, only two synonymous SNPs were identified within genes, in *HSP78* and *sterol 24-c-methyltransferase*. Although synonymous mutations are often not phenotypically relevant, they may affect translation by differences in codon usage. Both affected genes are of potential relevance to PMM resistance, as the *HSP78* gene in *L. donovani* is an important virulence factor, needed to suppress immune activation and escape from NO-mediated toxicity in macrophages [37], whereas sterol *24-c-methyltransferase* in *L. major* has been linked to mitochondrial function and virulence [38]. For both genes, a negative correlation was found between the alternate allele frequency and PMM resistance. The lower relative synonymous codon usage of the alternate codon in the *HSP78* gene (RSCU decrease from 1.439 to 1.115 for UCG → UCC) would indicate a higher expression to be linked to PMM resistance. A negative correlation was also found for *sterol 24-c-methyltransferase* alternate allele frequency and the PMM IC_50_ values. The reduced frequency of the alternate codon (codon UUC → UUU, RSCU decrease from 1.28 to 0.72) is anticipated to decrease the expression levels in favor of increased resistance. Genomic instability of *sterol 24-c-methyltransferase* has already been associated with amphotericin B resistance in *L. mexicana*, *L. major*, *L. donovani* and *L. infantum* [39,40]. In this study, however, no correlation could be demonstrated between the PMM resistance profiles of the different clones and their susceptibility towards amphotericin B. In a study with promastigote-selected PMM-resistant lines, no differential expression of *sterol 24-c-methyltransferase* was noted [40], but it should be mentioned that in these strains, PMM susceptibility of the amastigote stage remained unaltered [12,41], whereas selection on the amastigote level triggers a completely different outcome [9].

Despite stability of the PMM resistance upon parasite passage in mice or sand flies [9,42], suggesting a stable genomic modification, this comparison mostly identified gene CNVs correlating with resistance. Chromosomal somy changes could not be unequivocally correlated to the variable PMM susceptibility of the different clones. Due to the extremely high genome plasticity of *Leishmania*, somy changes should be cautiously interpreted, as they are readily affected by culture conditions, prolonged cultivation and the associated virulence changes [43]. Nonetheless, the CNV of 39 genes was found to be statistically associated with PMM resistance, where 33 were a result of gene expansions and 6 of gene losses. While 14 of these code for hypothetical proteins, the other 25 are known to be involved in transcription, translation and protein turn-over, virulence, mitochondrial function, signaling and vesicular trafficking. Although the exact mechanisms of action of PMM in *Leishmania* are yet to be elucidated, past research on experimentally selected resistant *L. donovani* already suggested that PMM not only impairs protein synthesis and modifies membrane fluidity and permeability, but also causes respiratory dysfunction with associated alterations of the mitochondrial membrane potential [11,32,35,40,44,45,46].

When only considering those genes with a copy number difference amplitude of at least 0.3 haploid genome copies, 11 genes could be identified, of which 6 have been annotated (i.e., *ADP/ATP mitochondrial carrier-like protein, ras-related protein RAB1, 60S ribosomal protein L6*, *proteasome regulatory non-ATP-ase subunit 3*, an *RNA-binding protein* and *ribosomal protein L1a*). ADP/ATP mitochondrial carrier-like proteins are known to transfer molecules across mitochondrial membranes [47] and could therefore potentially play a role in drug transport into a main target organelle. The Ras-related protein RAB1 is involved in vesicular trafficking and is a known regulator of secretory pathways of several important *Leishmania* virulence factors, including gp63 and acid phosphatase [48]. Proteasome regulatory non-ATP-ase subunit 3 is important in the regulation of proteosomal functions, of which inhibition hampers specific stages of morphological differentiation in various protozoan parasites, including *Leishmania* [49]. In *L. infantum*, proteasomal functions were shown to be essential for replication and intracellular survival of amastigotes in the host cell [50]. Its upregulation upon PMM selection thus supports the increased intracellular parasite fitness observed by our group and others [12,43,51]. The increased virulence may also relate to the detected CNVs for the virulence genes *major surface protease gp63* and *protein-tyrosine phosphatase 1-like protein* [52]. The identification of the *ribosomal protein L6* gene is also noteworthy as it has been shown to be involved in bacterial resistance against aminoglycosides in the past [53]. Moreover, its close homologue, the *60s ribosomal protein L23*, is known to be overexpressed in antimony-resistant parasites [54]. Additionally, *RNA-binding protein* overexpression has already been described in antimony-resistant strains [55], and could be linked to an increased parasite replication efficacy [56]. A negative correlation was found for the *ribosomal protein L1a* and *quiescin sulfhydryl oxidase*. Although not much information can be found in the literature on the role of both gene products in antileishmanial drug resistance, quiescin sulfhydryl oxidase has been associated with the induction of reduced cell proliferation and quiescence, for example in fibroblasts [57,58]. Its decreased expression may increase cell proliferation, which in theory fits the increased parasite fitness observed for PMM-resistant parasites [12]. Finally, the presence of several hypothetical proteins in which gene copy number alteration was potentially associated with PMM resistance reinforces the importance of characterizing unknown protein functions in *Leishmania*.

Other research groups have evaluated PMM resistance markers using genomic, metabolomic or transcriptomic comparisons [43,44,59]. Metabolic and lipidomic changes were shown to be strain-specific [43], and up until now, no suitable PMM resistance biomarkers have been identified. None of the post-selection variants that we found to be associated with PMM resistance were previously identified by earlier research as associated with *Leishmania*’s PMM resistance. However, this is not completely unexpected as the present study was the only one to have selected PMM resistance directly on intracellular amastigotes. Our results indicate the likely multifactorial nature of PMM resistance, which complicates the identification of a single genetic marker for application in clinical settings. In-depth research and validation of the identified genetic variations in a range of clinical isolates will be necessary to understand their biological importance and use as potential resistance markers.

## Figures and Tables

**Figure 1 microorganisms-09-01546-f001:**
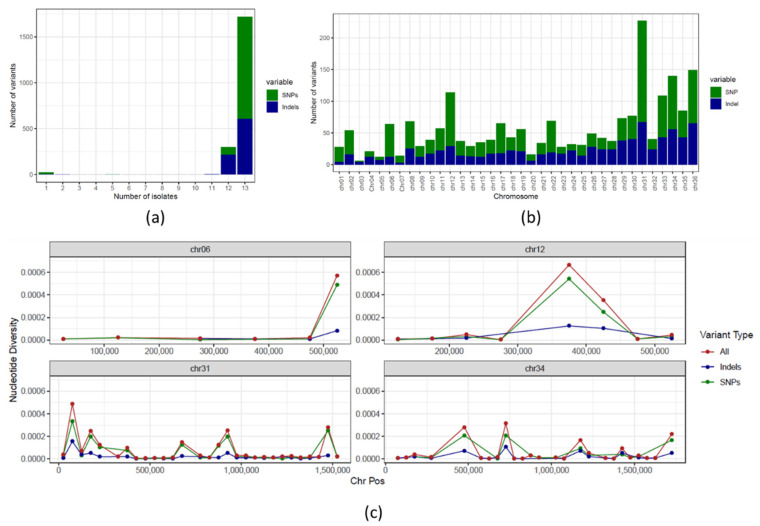
Genetic variability of the *L donovani* clones: (**a**) Number of SNV positions that are shared between exactly 1 to 12 clones and the polyclonal population. The Y and X axis represent, respectively, the number of variants and the exact number of isolates in which the variant position was observed. A total of 84% of the SNV positions were present in all clones and the non-cloned population. (**b**) Number of SNPs and indels in each chromosome. (**c**) Genetic diversity (π) in chromosomes 6, 12, 31 and 34 estimated using 50 kb windows. The Y and X axis represent, respectively, genetic diversity and chromosomal position. SNVs, indels and SNPs are represented in red, blue and green, respectively.

**Figure 2 microorganisms-09-01546-f002:**
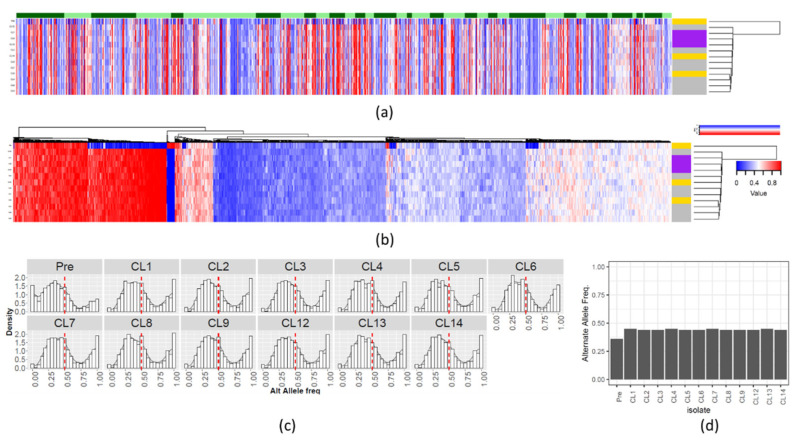
SNVs alternate allele frequency variability. Alternate allele frequency of each SNV position in the *L donovani* clones and preselection population. (**a**) SNVs ordered by chromosome position. The columns and lines correspond respectively to SNV positions and the various *L donovani* clones. The allele frequency of the alternate allele is represented in a scale from blue (low) to red (high). The dendrogram on the right represents the UPGMA clustering of the Manhattan distance of the alternate allele frequencies. The color strip beside the dendrogram represents the PMM resistance phenotype, where RI < 2, 2 > RI > 5 and RI > 5 are represented respectively, in yellow, grey and purple. The order of the isolates, from top to bottom, is: “preselection”, “CL 12”,“CL 1”,“CL 8”,“CL 13”,“CL 2”,“CL 14”,“CL 5”,“CL 7”,“CL 6”,“CL 4”,“CL 9” and “CL 13”. The alternating light/dark green color strip on the top represents the chromosomes 1–36. (**b**) The SNVs columns were reordered by UPGMA clustering of the Manhattan distance, instead of being ordered by chromosome position. (**c**) Genome-wide density plot of the alternate allele frequency distribution of SNVs in the 12 clones and the preselection population. The X and Y axis represent, respectively, the alternate allele frequency (from 0 to 1) and the allele frequency rate. The red dashed line represents an alternate allele frequency of 0.5, the expected value for disomic chromosomes. The observed distribution is suggestive of triploidy. (**d**) Median genome-wide allele frequency for each clone and the preselection population (Pre).

**Figure 3 microorganisms-09-01546-f003:**
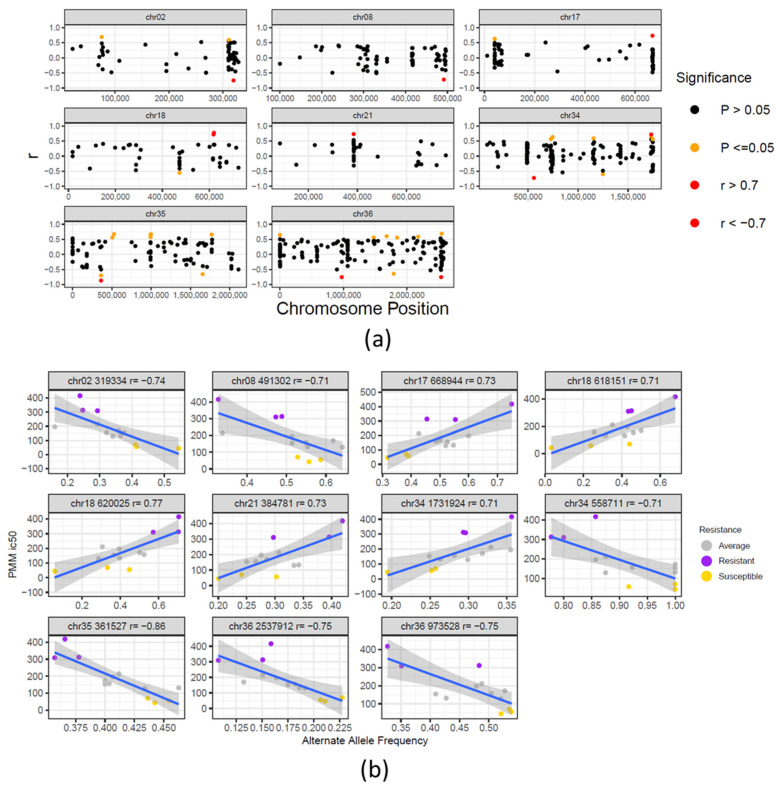
Correlation between SNVs and PMM resistance: (**a**) Chromosomal distribution and correlation of SNV allele frequency and PMM resistance. The X and Y axis correspond, respectively, to the chromosome position of each SNV and its correlation to the PMM resistance phenotype. Significant correlations are represented by orange dots. SNVs with strong significant positive or negative correlations (correlation coefficients r ≥ 0.7 and r ≤ −0.7) are represented by red dots. (**b**) Pearson correlation plots of the 11 significant SNVs with strong correlation to PMM resistance. The box title contains the SNV chromosome, position and correlation, r. The color of the dots reflects the level of resistance of the individual clones to PMM, where yellow, grey and purple correspond respectively to RI < 2, 2 < RI < 5 and RI > 5.

**Figure 4 microorganisms-09-01546-f004:**
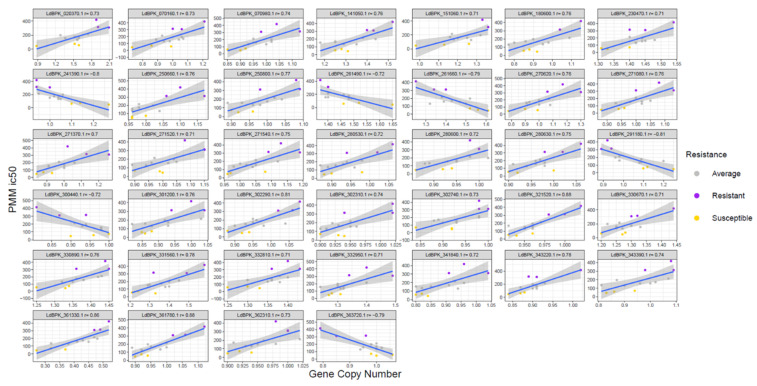
Correlation between gene CNVs and PMM resistance. Pearson correlation plots of the 39 significant genes with strong correlations to PMM resistance. The title for each box contains the TriTrypDB gene ID and r correlation. The color of the dots reflects the clone’s resistance to PMM, where yellow, grey and purple correspond respectively to RI < 2, 2 < RI < 5 and RI > 5.

**Table 1 microorganisms-09-01546-t001:** Intracellular amastigote IC_50_ values of the pre-selection population and the 12 different post-selection clones against paromomycin (PMM) and amphotericin B (AmB). Results are expressed as the average IC_50_ (in µM) ± the standard error of the mean (SEM). The clones that were considered resistant (resistance index (RI) > 5) are indicated in bold. Those considered susceptible (RI < 2) are indicated in italics. All reported IC_50_ values are the result of at least two independent experiments run in duplicate.

Strain	PMM IC_50_ (µM) ^1^	AmB IC_50_ (nM)
Wild-type (preselection)	*45.0 ± 5.6*	9.7 ± 0.2
Clone 1	**417.4 ± 15.1**	20.9 ± 6.6
Clone 2	196.8 ±11.2	15.4 ± 2.0
Clone 3	213.0 ± 7.3	19.6 ± 4.5
Clone 4	157.0 ± 9.6	22.4 ± 6.1
Clone 5	129.7 ± 13.1	10.5 ± 1.0
Clone 6	*57.1 ± 6.8*	11.8 ± 2.1
Clone 7	154.0 ± 14.9	22.0 ± 5.1
Clone 8	**313.1 ± 14.3**	15.3 ± 0.9
Clone 9	132.5 ± 9.0	13.8 ± 0.8
Clone 12	171.3 ± 5.7	13.7 ± 1.7
Clone 13	**310.2 ± 11.8**	17.0 ± 3.5
Clone 14	*71.2 ± 2.8*	ND

^1^ These results have already been published elsewhere [9]. ND: Not done.

**Table 2 microorganisms-09-01546-t002:** Genes potentially associated with PMM resistance. Correlation between gene copy number and PMM resistance. Only genes for which the correlation, r, was ≥0.7 or ≤−0.7 and the correlation *p*-values were lower than 0.05 are reported. GeneID: TriTrypDB *L donovani* gene ID; Chr: chromosome number; Pos 1: gene initial coordinate; Pos 2: gene final coordinate; Correlation: Gene CN and PMM resistance correlation; *p*: Correlation *p*-value; Max: Maximum gene CN in all clone/population; Min: Minimum gene CN in all clone/population; Max–Min: the difference between the Max and Min gene CNV values.

Gene ID	Chr	Pos1	Pos2	Gene Annotation	Correlation	*p*	Max	Min	Max − Min
LdBPK_020370.1	chr02	171561	171713	hypothetical protein unknown function	0.73	4.96 × 10^−3^	2.09	0.88	1.21
LdBPK_070160.1	chr07	67067	67567	hypothetical protein conserved	0.73	4.54 × 10^−3^	1.21	0.73	0.48
LdBPK_070980.1	chr07	404916	406232	hypothetical protein conserved	0.74	3.96 × 10^−3^	1.13	0.85	0.28
LdBPK_141050.1	chr14	424294	425424	ADP/ATP mitochondrial carrier-like protein	0.76	2.45 × 10^−3^	1.52	1.18	0.34
LdBPK_180600.1	chr18	242208	242483	hypothetical protein	0.76	2.47 × 10^−3^	1.15	0.77	0.38
LdBPK_230470.1	chr23	162446	163672	aldose 1-epimerase-like protein	0.71	6.97 × 10^−3^	1.54	1.31	0.23
LdBPK_250660.1	chr25	217458	219455	predicted zinc finger protein	0.76	2.84 × 10^−3^	1.17	0.96	0.21
LdBPK_250800.1	chr25	276286	277551	hypothetical protein unknown function	0.77	2.10 × 10^−3^	1.09	0.89	0.2
LdBPK_270620.1	chr27	239023	239625	ras-related protein RAB1A putative	0.76	2.58 × 10^−3^	1.3	0.78	0.52
LdBPK_271080.1	chr27	441116	442363	RING-H2 zinc finger putative	0.76	2.47 × 10^−3^	1.13	0.88	0.25
LdBPK_271370.1	chr27	539921	540925	proteasome regulatory non-ATP-ase subunit 3 putative	0.70	7.57 × 10^−3^	1.26	0.84	0.42
LdBPK_271520.1	chr27	591692	592333	Eukaryotic translation initiation factor 4E−1	0.71	6.85 × 10^−3^	1.15	0.89	0.26
LdBPK_271540.1	chr27	607146	609893	BRO1-like domain/ALIX V-shaped domain binding to HIV putative	0.75	3.40 × 10^−3^	1.19	0.96	0.23
LdBPK_280530.1	chr28	180433	182571	TPR repeat putative	0.72	5.12 × 10^−3^	1.06	0.87	0.19
LdBPK_280600.1	chr28	209904	211604	major surface protease gp63 putative	0.72	5.67 × 10^−3^	1.02	0.86	0.16
LdBPK_280630.1	chr28	222510	224015	hypothetical protein unknown function	0.75	3.31 × 10^−3^	1.06	0.9	0.16
LdBPK_301200.1	chr30	387893	388444	hypothetical protein conserved	0.76	2.82 × 10^−3^	1.04	0.82	0.22
LdBPK_302290.1	chr30	855859	857973	tubulin-tyrosine ligase-like protein	0.81	8.65 × 10^−4^	1.08	0.88	0.2
LdBPK_302310.1	chr30	864017	867931	hypothetical protein conserved	0.74	3.75 × 10^−3^	1.02	0.9	0.12
LdBPK_302740.1	chr30	1031040	1032266	TPR domain protein conserved	0.73	4.95 × 10^−3^	1	0.84	0.16
LdBPK_321520.1	chr32	572415	582050	phosphatidylinositol 3-related kinase putative	0.88	7.93 × 10^−5^	1.02	0.93	0.09
LdBPK_330670.1	chr33	209670	210923	intraflagellar transport protein 57/55 putative	0.71	6.32 × 10^−3^	1.44	1.2	0.24
LdBPK_330890.1	chr33	292053	294803	hypothetical protein unknown function	0.76	2.47 × 10^−3^	1.45	1.25	0.2
LdBPK_151060.1	chr33	429431	429940	60S ribosomal protein L6 putative	0.71	6.78 × 10^−3^	1.36	0.98	0.38
LdBPK_331560.1	chr33	594808	595194	RNA-binding protein putative	0.78	1.70 × 10^−3^	1.58	1.21	0.37
LdBPK_332810.1	chr33	1086018	1089704	hypothetical protein conserved	0.71	7.02 × 10^−3^	1.43	1.25	0.18
LdBPK_332950.1	chr33	1165769	1167190	transcription elongation factor-like protein	0.71	6.23 × 10^−3^	1.49	1.24	0.25
LdBPK_341840.1	chr34	802817	803974	protein kinase putative	0.72	5.11 × 10^−3^	1.04	0.8	0.24
LdBPK_343220.1	chr34	1362570	1364477	DNA topoisomerase IB large subunit	0.78	1.48 × 10^−3^	1.02	0.84	0.18
LdBPK_343390.1	chr34	1417106	1417627	Complex 1 protein (LYR family) putative	0.74	4.04 × 10^−3^	1.09	0.81	0.28
LdBPK_361330.1	chr36	475562	475984	hypothetical protein (fragment)	0.86	1.79 × 10^−4^	0.52	0.27	0.25
LdBPK_361780.1	chr36	673858	675498	hypothetical protein conserved	0.88	8.22 × 10^−5^	1.12	0.89	0.23
LdBPK_362310.1	chr36	878592	880163	protein-tyrosine phosphatase 1-like protein	0.73	4.69 × 10^−3^	1.02	0.9	0.12
LdBPK_241390.1	chr24	496214	498880	hypothetical protein conserved	−0.80	1.13 × 10^−3^	1.27	0.94	0.33
LdBPK_261490.1	chr26	541997	546778	hypothetical protein unknown function	−0.72	5.61 × 10^−3^	1.65	1.37	0.28
LdBPK_261660.1	chr26	599794	600579	hypothetical protein conserved	−0.79	1.19 × 10^−3^	1.61	1.25	0.36
LdBPK_291180.1	chr29	429699	429965	ribosomal protein L1aputative (fragment)	−0.81	7.78 × 10^−4^	1.25	0.89	0.36
LdBPK_300440.1	chr30	142693	144375	quiescin sulfhydryl oxidase putative	−0.72	5.96 × 10^−3^	1	0.81	0.19
LdBPK_363720.1	chr36	1394227	1395405	phytoene synthase putative	−0.79	1.33 × 10^−3^	1.06	0.79	0.27

## Data Availability

Short-read data has been submitted to the National Center for Biotechnology Information (NCBI) (BioProject: PRJNA739251). SRA records will be accessible at https://www.ncbi.nlm.nih.gov/sra/PRJNA739251. All other data is available at Figshare (https://figshare.com/authors/Daniel_Jeffares/649018; access on 19 July 2021).

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
