# Peer review of "Experimental Selection of Paromomycin Resistance in Leishmania donovani Amastigotes Induces Variable Genomic Polymorphisms"

_microorganisms, 2021, doi:10.3390/microorganisms9081546_

Round 1

Reviewer 1 Report

This is an interesting and nicely written manuscript that approaches an important  area of research involving drug resistance in Leishmania.  The authors clearly present their work in a well organized way.  The use of the drug resistance clones and extensive analyses of the genomic data allows for reasonably strong conclusions about 25 genes potentially involved in resistance to paromomycin; their selection of 11 genes with a copy number difference amplitude clearly helps focus on the multifactorial nature of paromomycin resistance. Their discussion and conclusions are well done and very interesting and give the reader much to think about concerning what should be done next.  My only comment involves use of abbreviations without first defining them; for example lines 48 (define SNVs first time used); lines 74 and 75 (define SNPs and INDELS); line 96 define UPGMA.

Author Response

See submitted file

Reviewer 2 Report

I found the work of  Hendrickx et al. important and timely. My comments and suggestions are mainly cosmetic, but I believe the way this story is presented must be improved and the text must be revised for clarity. 

1)  Introduction. I suggest to add a sentence about Trypanosomatidae in general, citing, f.e. doi:10.1017/S0031182018000951. 

2) Materials and Methods. I noticed numerous typos and inconsistences.

2a) Please unify the way you refer to manufacturers. Sometimes cities and countries are mentioned (ln. 53), sometimes they are not (elsewhere). 

2b) NEB is ln. 65 must be spelled out. 

2c) There are 2 dots in ln. 66. 

2d) TriTrypDB must be referenced doi:10.1093/nar/gkp851. 

3) Results Results and discussion. There are numerous typos creating an impression of sloppiness. I will provide just some examples, but the whole text must be checked and revised. 

3a) In ln. 125 (Table 1) is written as (table 1).   

3b) Please reformat the text to avoid half-blank pages - pg. 3 and pg. 5. 

3c) Please provide a citation for the statement in lns. 300-301. Also, it may be important to discuss it in the context of doi:10.1016/j.pt.2020.10.001. 

4) Funding. ln. 371. "was supported" x2

5) References. These must be unified and put in the correct format. Why all  words in some titles are capitalized and in some they are not? Why some Journal names are abbreviated and some aren't? Why there are periods between words in some Journal titles? #47 - Acta Trop. (not trop.) Also, please put spaces between words in Journal titles.  

Author Response

See submitted file

Reviewer 3 Report

This is a well written manuscript.  I am not qualified to comment on the methods involved in manipulation of the genetic data from this study - but as a non-specialist I followed the methods.

Author Response

See submitted file